# Assessment, Distribution and Regional Geochemical Baseline of Heavy Metals in Soils of Densely Populated Area: A Case Study

**DOI:** 10.3390/ijerph17072269

**Published:** 2020-03-27

**Authors:** Song Chen, Cancan Wu, Shenshen Hong, Qianqian Chen

**Affiliations:** 1School of Resources and Civil Engineering, Suzhou University, Suzhou 234000, China; szxywcc@163.com (C.W.); HSS990706@163.com (S.H.); 15755653557@163.com (Q.C.); 2National Engineering Research Center of Coal Mine Water Hazard Controlling, Suzhou 234000, China; 3Key Laboratory of Mine Water Resources Utilization of Anhui Education Department, Suzhou 234000, China

**Keywords:** assessment, distribution, geochemical baseline, heavy metal, densely populated area

## Abstract

To understand the content, pollution, distribution and source and to establish a geochemical baseline of heavy metal elements in soil under the influence of high-density population, the concentrations of heavy metal elements Cr, Mn, Co, Ni, Cu, Zn, As, Cd, Hg, Pb and Fe were determined in 23 soil samples in Suzhou University, and geo-accumulation index, enrichment factor, principal component analysis, spatial analysis and regression analysis were completed. The results showed the following: The elements Cu and As were slightly polluted, while the other heavy metal elements were not. The elements Cd, Cu, Ni and As in soils were mainly caused by agricultural activities of chemical fertilizer, whereas the elements Zn and Hg were impacted by the chemicals and batteries. The heavy metal elements in the north were lower than in the south of the campus, as a whole. The enrichment of elements Cu, As and Cd was caused by the east–west river on the campus, and the enrichment of the elements Mn, Ni and Zn was induced by the reservoir. Biochemical experiments and vehicle parking influenced the spatial enrichment of Cr, Co and Pb, while domestic waste led to the spatial differentiation of Hg concentrations. The regression curve between heavy metal elements and Fe was established, and the background values of the heavy metals Cr, Mn, Co, Ni, Cu, Zn, As, Cd, Hg and Pb are 50.90, 489.37, 11.76, 37.74, 55.70, 58.22, 20.07, 0.09, 0.08 and 24.13 mg/kg, respectively.

## 1. Introduction

Soil is a natural resource with a high intrinsic value for ecosystems and humans that formed the environment with other factors, such as water, air, rock and organisms; therefore, it must be conserved and protected. In general, the content and distribution of chemical elements in soil depend on the composition of the parent rock, the soil-forming process, climate, topography and land use [1,2]. Thus, the chemical elements present in a soil can be lithogenic, pedogenic, and anthropogenic, especially the heavy metal elements contents in soils seriously changed by human activity (industry, household garbage and agriculture) [3,4].

How to eliminate the human impact and obtain the background value of heavy metals in soil are very important for the evaluation of soil pollution. Relative cumulative frequency and regression analysis are mainly used to constitute the heavy metal geochemical baseline in soils [5,6]. The relative cumulative frequency method was established in the 1990s, and then improved by domestic and overseas scholar. Regression analysis is good for revealed the geochemical natural background concentrations of elements without any anthropogenic enrichment.

With only trace levels in soils, heavy metals are well-established to be detrimental to human health and environment. Excessive heavy metal content will directly harm human health; it can even affect the growth and reproduction of animals and plants, and then affect the survival of organisms [7,8]. The content of heavy metals in the soils is relatively low in the natural state, but they tend to accumulate from human activities. The common heavy metals Cr, Mn, Co, Ni, Cu, Zn, As, Cd, Hg and Pb are not only ubiquitous in the soil, but also easy to be released by human activities. Therefore, it is beneficial for soil evaluation and human health to carry out research on these heavy metal elements.

In order to better evaluate the harm of heavy metals in soils to humans and the natural environment. Several attempts have been made to separate between anthropogenic and natural contributions on metal contents in soils. Previous studies showed these methods such as statistic evaluation of the outliers, regressions between metals and Al and Fe, and spatial analyses are good to identify heavy mental sources in soils, as well as to establish regional geochemical baseline [9,10]. Moreover, geo-accumulation and the enrichment factor are the common assessment methods of heavy metal in soils [11,12].

Many studies have focused on the content, evaluation, sources and distribution of heavy metal elements in soil in cities, villages and mountainous areas, but the related research in small-scale densely populated areas is less. In China, colleges and universities are typical places in small-scale densely populated areas. In this study, we report the latest heavy metal elements in soils from different regions in university campuses. The main research objectives are (1) determining the degrees of heavy metal pollution; (2) analyzing the spatial distribution characteristics and sources of heavy metal elements in soil; and (3) constituting the regional baseline geochemical and calculating the background of heavy metal.

## 2. Materials and Methods

### 2.1. Study Area

In order to facilitate the implementation of our research work, Suzhou University campus was selected as the research area. The campus was founded in 2004, with a permanent population of about 15,000 people, covering an area of about six hundred thousand square meters. The study area is located in the east of Suzhou city, in Northern Anhui Province, China (Figure 1), with longitude ranging from 117.0796–117.0878° and latitudes from 33.6341–33.6449°. In the north and east of the campus are a middle school and a provincial road, respectively, and the south and west are surrounded by farmlands. Suzhou city has a large population and belongs to the warm temperate semi-humid monsoon climate area. The economy in the area is mainly agricultural plating and coal industry. There is an east–west river and a reservoir in the campus. Living areas are in the north of the river and include students’ dormitories, canteens and sports grounds. To the south of the river is the teaching area, including the experimental building, teaching building and library.

### 2.2. Sampling Test and Research Method

A total of 23 soil samples were collected in October 2019, and the site coordinates were noted (Figure 1). Soil samples were collected from the outer surface, i.e., 5–15 cm depth, by using a plastic spatula, to for avoid the influence by metal tools. Then, samples were packed into polyethylene bags and returned to the laboratory. The soils were then air-dried at room temperature and pressed through a 2 mm stainless-steel mesh screen, to remove debris of plants and stones. About 100 g of soils were removed by using the quartered method, and then they were ground and pushed through another 200-mesh nylon sieve for analysis. Samples pellets were prepared for analysis of heavy metal elements by X-ray fluorescence (XRF) (Explorer 9000SDD) spectrometry (INOS China Co., Ltd, Shanghai City, China), a backing of boric acid in collapsible aluminum cups and pressing at 30 t of pressure. The ten heavy metal elements, Cr, Mn, Co, Ni, Cu, Zn, As, Cd, Hg and Pb, and the major element Fe, were measured. The experiment was carried out in the key laboratory of mine water resources utilization of Anhui education department. National standard sediment sample (GBW07307) was analyzed simultaneously for calibration, and the relative standard derivation was less than 10%.

Geo-accumulation index (*I_geo_*) and Enrichment factor (*EF*) methods were selected to assess the pollution degree of heavy metal in soils. The *I_geo_* enables the assessment of contamination degrees by comparing the current and pre-industrial concentrations [11], and it is calculated via Equation (1):(1)$$I_{\mathrm{geo}}=\log_{2}C_{n}÷1.5B_{n}$$
where *C_n_* is the measured concentration of the element in the samples, and *B_n_* is the background or pristine value of the element. Previous studies showed the measurement of I_geo_ can be subdivided into 5 degrees: <0—unpolluted; 0–1—light pollution; 1–3—moderate pollution; 3–5—heavy pollution; >5—serious pollution [11].

The *EF* is calculated via Equation (2):(2)$$EF=M_{x}\times Mr_{b}÷M_{b}\times Mr_{x}$$
where *M_x_* and *Mr_x_* are the sample concentrations of the heavy metal and reference element, while *M_b_* and *Mr_b_* are their concentrations in a suitable background or baseline reference material [12].The *EF* also could be subdivided into 5 degrees in previous studies: <2—unpolluted; 2–5—light pollution; 5–20—moderate pollution; 20–40—heavy pollution; >40—serious pollution [12]. The *EF* method normalizes the measured heavy metal content with respect to a sample reference metal such as Fe or Al [13].

In the study of environmental geochemistry, statistical analyses, such as Principal Component Analysis and Cluster Analysis, are often used to reveal the relationship between elements or parameters. In particular, the Principle Component Analysis is an efficient way of displaying complex relationships among many variables and their roles, and the Principal Component Analysis was also used in this study, to explain the controlling factors of heavy metal content in soil. Relative cumulative frequency and regression analysis are mainly used to constitute the heavy metal geochemical baseline in soils. Regression analysis is good for revealing the geochemical natural background concentrations of elements without any anthropogenic enrichment [6,14]. The reference element, a fundamental or conservative tracer of the natural metal-bound phases, is assumed to have a uniform flux that forms the coastal bedrock. Various elements have been proposed in the literature to be the potential conservative reference element, such as Al and Fe [6,14]. All the statistical analyses of the data were performed by Minitab software (version 15), and the contour map of heavy metal spatial distribution was completed by Surfer software (version 8.0) (Kingsoft company, Beijing City, China).

## 3. Results and Discussion

### 3.1. Basic Description of Data

The basic statistical results of heavy metal in soils in study area and the background value in China are listed in Table 1. It can be seen from Table 1 that the average content of Fe is close to the soil background value of China (2.92%), with the value be 2.71% [15]. The coefficient of variation (CV) could reflect the dispersion degree of the sample in space. As can be seen, the CV of element Fe is the lowest, which is 0.09%. Moreover, the average content of element Fe is 2.71 mg/kg, the same as the Median value. It also shows that the distribution of Fe content in soil is stable. In detail, the average contents of Ni, Cu, As, Cd and Hg in the sample are higher than the soil background values of China [15], which are 40.06, 60.75, 25.03, 0.10 and 0.12 mg/kg, respectively. The average contents of other heavy metals are slightly lower than the soil background values. The CV of element Hg is highest, at 132.94%; the second is 82.80%, and Cu is the third, at 32.68. The CVs of all other heavy metals are smaller, less than 30%.

The average value of heavy metal exceeds the background value, which indicates that heavy metal in soil may be enriched by other sources. The maximum sample points of the heavy metal are analyzed and summarized as follows: (1) The concentrations of Cu, As and Cd in sample 17 are the highest, which are 101.37, 85.65 and 0.14 mg/kg; (2) the concentrations of Cr, Co, Pb and Fe in sample 2 are highest, which are 61.6, 12.43, 27.43 mg/kg and 3.39%; (3) the highest concentrations of Mn, Ni, Zn and Hg are distributed in sample 22, sample 14, sample 19 and sample 4, respectively, with the maximum value 698.07, 54.26, 116.22 and 0.55 mg/kg. It should be noted that sample 17’s site is located under the crossing bridge (Jiayan Bridge) between the main road and the river on the campus, and sample 2 was collected at the entrance of the biochemical laboratory building. Sample 22, sample 14, sample 19 and sample 4 were respectively located on both sides of the reservoir, under the bridge and near the canteen, respectively.

### 3.2. Environmental Risk Evaluation

The *I_geo_* and *EF* are the common methods for heavy metal assessment in soils. As a "proxy" for the clay content, the element Fe is also considered to act as the reference element to calculate the *EF* [13]. The CV of Fe in Table 1 shows the contents of the Fe in soils are stable and the average value of Fe content is close to the background value of China. These characteristics, such as stability and lack of obvious artificial sources, are the standards of selection of reference elements; thus, the element Fe is selected as the reference element in this study, to calculate the *EF* of heavy metals.

According to the Equations (1) and (2), the *I_geo_* and *EF* of heavy metal elements in soils were calculated, and the results of the *I_geo_* and *EF* are also presented in Table 1. In Section 2.2 of this paper, the classification standards of *I_geo_* and *EF* are introduced; according to the standard of *I_geo_* [11], the elements Cu and As in the soil are slightly polluted, with the I_geo_ index 0.77 and 0.30. Moreover, other heavy metal elements are unpolluted, with the I_geo_ index being less than zero. Compare with the *EF* standard [12], the elements Cu and As in the soil are slightly polluted, with the *EF* 2.92 and 2.50, and other heavy metal elements are almost unpolluted. The evaluation results of heavy metal pollution by *EF* and *I_geo_* are basically the same. In conclusion, the heavy metal elements Cu and As in the soil of this area are slightly polluted, and other heavy metals are not polluted.

### 3.3. Statistical Analysis

As mentioned earlier, the Principle Component Analysis is an efficient way of displaying complex relationships among many variables and their roles [16]. In order to further reveal the relationship between heavy metal elements in soils, the Principle Component Analysis is conducted for the heavy metal concentrations. The rotated Principal Component Loadings are given in Table 2 and Figure 2.

Three Principal Components emerged with the eigenvalue all bigger than 1, which explained more than 76.48% of cumulative variance. The PC1 represents Cd, Cu, Ni and As, with 34.07% of the total variance, while the PC2 includes Fe, Co, Pb, Cr and Mn, with 28.41% of the total variance. The elements Zn and Hg are controlled by PC3, with 14.01% of the variance. The score plot of the three PCs are shown in Figure 2, and it is characterized by the following: (1) The elements Cu, Ni, As and Cd are considered a group for the higher values in PC1; (2) the elements Fe, Pb, Co and Mn are considered a group for the similar behavior; (3) the elements Zn and Hg are separate, with their representative high value in PC3; (4) the element Cr has a higher value in PC2 and PC1.

Combined with the previous results of elements, Cu and As in soils are lightly polluted, though the PC1 could be one of the factors contributing to the heavy metal concentrations in the soils of the study area. Previous studies show that alloy, fertilizer, coating, mining and smelting could discharge Cu, Ni, As and Cd at the same time, which leads to the enrichment of heavy metal in soils, water bodies and street dust [17]. The University campus was farmland before the construction, and there were no industrial activities such, as alloy, coating and mining. Moreover, it is speculated that the Cd, Cu, Ni and As in soils are mainly caused by agricultural activities of chemical fertilizer; thus, PC1 represents the impact of agricultural activities. The element Fe is lacking obvious artificial sources, and the elements Co, Pb, Cr and Mn are unpolluted; PC2 represents the natural Pedogenesis in the area. Similarly, the main human activities that can release Zn and Hg at the same time are chemicals, batteries, fertilizers and mining. Thus, PC3 could represent the chemicals and batteries for the higher value of Zn and Hg in PC3.

### 3.4. Distribution of Heavy Metals

The contour map of spatial distribution of heavy metal elements can reveal the distribution characteristics, spatial migration and sources of heavy metals to a certain extent [18,19]. The contour map of spatial distribution of heavy metals in the study area was plotted by Surfer software (Figure 3), and the Kriging interpolation method was used to calculate the unmeasured area. As a whole, the content of heavy metal elements in the living area in the north of the campus is relatively low, while the content in the teaching area in the south is relatively high, except for the element Hg. The element Hg has different characteristics from other heavy metal elements; it had a higher value in the living area and a lower value in the teaching area. It is concluded that the discharge of domestic waste, including waste batteries, may lead to the enrichment of Hg in the soil of the northern living area.

In addition, the elements Cu, As and Cd are enriched in the river bridge in the campus. Combined with the results of the previous principal component analysis, these elements may be enriched in the soil due to the fertilization process. Therefore, the enrichment of Cu, As and Cd in the campus is related to the surrounding farmland, which may be brought by the river runoff from upstream. Although the elements Mn, Ni and Zn are enriched around the reservoir in campus, the overall content of Mn, Ni and Zn in the study area is not high, and the impact of human activities is weak. This enrichment may reveal a local sink formed by the influence of topography and rainwater runoff in the campus. In addition, the elements Cr, Co and Pb are enriched in the experimental building and teaching building area, which should be affected by biochemical experiments and vehicle parking.

### 3.5. Regional Geochemical Baseline of Heavy Metal

The regression analysis is often used to construct geochemical baseline, for it can eliminate the influence of anthropogenic and reveal the geochemical natural background concentrations of elements [20]. Moreover, the selection of the reference element is very important before the regression analysis. The reference element is the basic or conservative tracer of the natural metal combination, and a variety of elements are proposed as potential conservative reference elements, such as Al, Fe, Li, Sc and so on. Among these elements, the elements Al and Fe are the most frequently used elements, for they are easily detected.

In addition, Section 3.1 in the paper mentions that the content and distribution of element Fe are stable and not affected by the environment. Therefore, Fe is selected as the reference element to establish the regional geochemical baseline of heavy metal in soils. The established regional geochemical baseline equation of heavy metal is shown in Figure 4. The regression equations between all ten heavy metals and Fe, the 95% confidence interval and prediction interval can be seen in Figure 4.

At present, the median of elements in the samples are very close to the average of all 10 heavy metal elements, indicating that the statistical distribution patterns of different heavy metal in samples are similar. What is also evidenced is the CV of heavy metals in Table 1. All 23 soil samples can be obtained from single and uniform parent materials, and their relatively small variability and narrow range between the minimum and maximum values of all elements also confirm this. In addition, this view is supported by the scenario that almost all soil samples in Figure 4 are within the 95% prediction range.

Previous studies show that the background concentration is determined by analyzing samples which are not affected by human activities, or at least affected by human activities, or according to the regression analysis calculation of standardized elements, or calculated by regression analysis, which based on the normalized element to select the upper 95% confidence interval of the linear regression between reference element and element of interest [21,22].

All data points which fall inside the 95% confidence band can be designated as natural sediments, without any contamination, while points above this area can be considered to be sediments with heavy metal accumulated from anthropogenic source. Thus, the regional background concentrations of heavy metal elements could be calculated by the average values of the natural sediment samples. According to the regression curve (Figure 4), in the 95% confidence interval of the regression curve, the number of sample points of heavy metal elements Cr, Mn, Co, Ni, Cu, Zn, As, Cd, Hg and Pb is 10, 11, 10, 7, 11, 16, 15, 9, 9 and 9, respectively. These sample points can be considered to be natural soil sample points without anthropogenic origin, and the average value of these sample points is the background value of heavy metal in the campus. Thus, the background values of the heavy metals Cr, Mn, Co, Ni, Cu, Zn, As, Cd, Hg and Pb are 50.90, 489.37, 11.76, 37.74, 55.70, 58.22, 20.07, 0.09, 0.08 and 24.13 mg/kg, respectively.

## 4. Conclusions

The average values of Ni, Cu, As, Cd and Hg contents were higher than the background values of Chinese soil, and the CV of Hg and As were bigger. The elements Cu and As were slightly polluted based on the evaluation methods of *I_geo_* and *EF*, while other heavy metal elements were not. Principal component analysis showed that the elements Cd, Cu, Ni and As in soils were mainly caused by agricultural activities of chemical fertilizer, and the elements Co, Pb, Cr and Mn represented the natural Pedogenesis in the area, whereas the elements Zn and Hg were impacted by the chemicals and batteries.

With the exclusion of Hg, the other nine heavy metal elements had the lower concentrations in the living area in the north of the campus, while the values in the teaching area in the south were relatively higher. The enrichment of Hg in soil could be induced by the domestic waste, including waste batteries. The elements of Cu, As and Cd were enriched in the river bridge in the campus, which were related to the surrounding farmland. The reservoir in the campus, as a lock sink formed by the topography, enriched the element Mn, Ni and Zn. The spatial enrichments of Cr, Co and Pb were caused by biochemical experiments and vehicle parking. 

The regression curves between heavy metal elements and Fe were established in the 95% confidence interval; the number sample points of heavy metal elements Cr, Mn, Co, Ni, Cu, Zn, As, Cd, Hg and Pb was 10, 11, 10, 7, 11, 16, 15, 9, 9 and 9, respectively. The background values of the heavy metals Cr, Mn, Co, Ni, Cu, Zn, As, Cd, Hg and Pb were 50.90, 489.37, 11.76, 37.74, 55.70, 58.22, 20.07, 0.09, 0.08 and 24.13 mg/kg, respectively.

## Figures and Tables

**Figure 1 ijerph-17-02269-f001:**
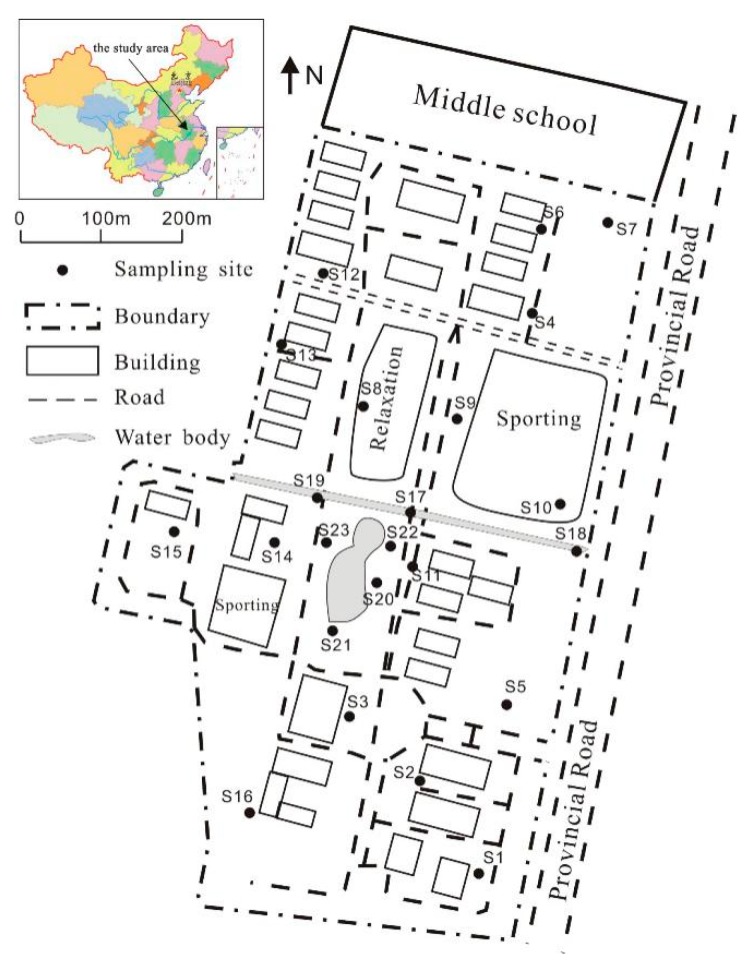
Study area and sampling sites.

**Figure 2 ijerph-17-02269-f002:**
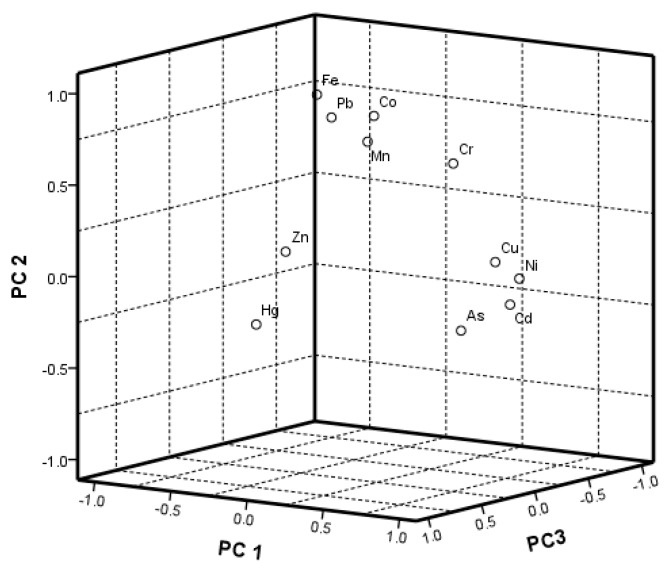
Loading plots of principal component factors of heavy metals in the study area.

**Figure 3 ijerph-17-02269-f003:**
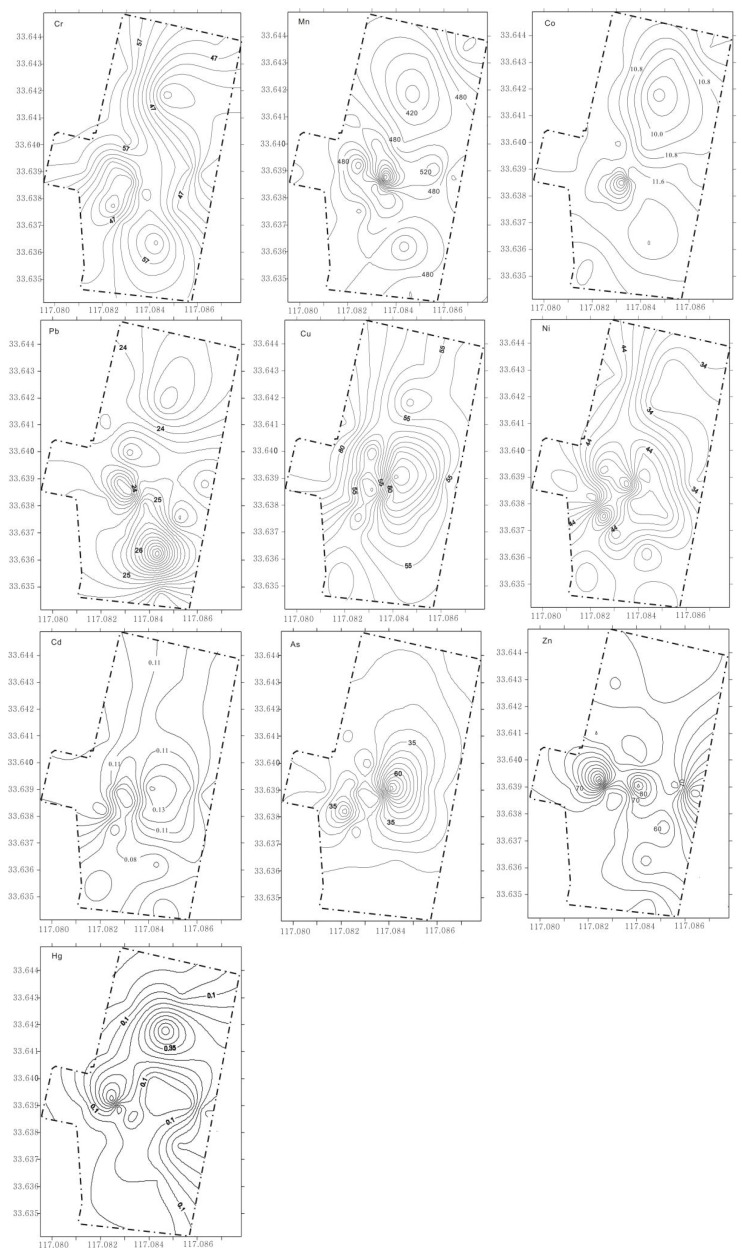
Contour map of spatial distribution of heavy metal in soils in the study area.

**Figure 4 ijerph-17-02269-f004:**
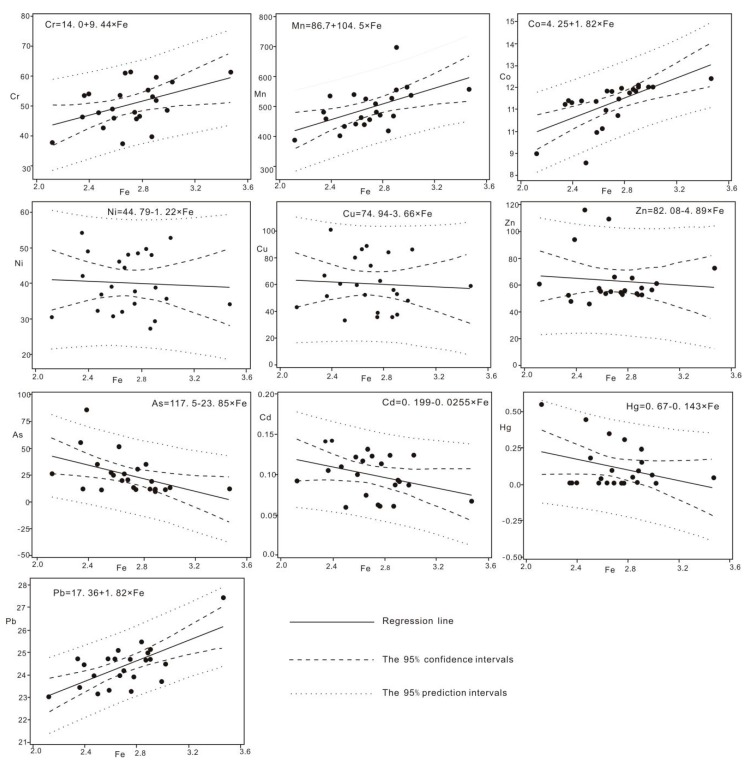
The linear regression models between the heavy metal elements and Fe of soils in the study area (*n* = 23).

**Table 1 ijerph-17-02269-t001:** Heavy metal (mg/kg) and Fe (%) concentrations of soil in the study area.

Element	Cr	Mn	Co	Ni	Cu	Zn	As	Cd	Hg	Pb	Fe
No.-H	S-2	S-22	S-2	S-14	S-17	S-19	S-17	S-17	S-4	S-2	S-2
Average	50.56	492.43	11.28	40.06	60.75	63.12	25.03	0.10	0.12	24.39	2.71
Min	37.61	385.19	8.57	27.28	33.39	46.1	10.96	0.06	0.01	23.02	2.20
Max	61.6	698.07	12.43	54.26	101.37	116.22	85.65	0.14	0.55	27.43	3.39
Median	49.2	478.06	11.48	38.71	58.03	56.24	19.72	0.10	0.05	24.52	2.71
CV(%)	14.30	13.90	8.85	20.49	32.68	29.19	72.80	26.63	132.94	3.93	0.09
BC	61	583	12.7	26.9	22.6	72.4	11.2	0.097	0.065	26	2.92
I_geo_	−0.87	−0.84	−0.76	−0.04	0.77	−0.83	0.30	−0.60	−1.12	−0.68	-
EF	0.89	0.91	0.96	1.62	2.92	0.95	2.50	1.12	2.06	1.02	-

Note: No.-H—highest content sampling No.; Min—minimum content; Max—maximum content; CV—coefficient of variation; BC—background values of China; I_geo_—geo-accumulation index; EF—enrichment factor.

**Table 2 ijerph-17-02269-t002:** Variance explained and component matrixes for heavy contents (*n* = 23).

Parameter	PC1	PC2	PC3
Cd	0.93	0.01	0.00
Cu	0.93	0.23	0.02
Ni	0.82	0.17	−0.30
As	0.78	−0.20	0.31
Fe	−0.22	0.87	−0.27
Co	0.22	0.84	−0.17
Pb	0.12	0.78	0.10
Cr	0.52	0.70	−0.37
Mn	−0.06	0.69	−0.41
Zn	0.16	−0.01	0.90
Hg	−0.31	−0.46	0.74
Eigenvalue	3.75	3.13	1.54
% of variance	34.07	28.41	14.01
Cumulative % variance	34.07	62.47	76.48

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
