# Peer review of "Assessment, Distribution and Regional Geochemical Baseline of Heavy Metals in Soils of Densely Populated Area: A Case Study"

_ijerph, 2020, doi:10.3390/ijerph17072269_

Round 1

Reviewer 1 Report

The authors have investigated the concentrations of heavy metals and asessed the risk of pollutions in a densely populated area. It is useful and informative for the environment management. Overall, the MS was written sound. The introduction and methods were presented clearly. The results were analysed statistically. The conclusions were based on the findings. Howerer, the discussion was weak. It is recommended to be published after minor revision.

  1. strengthen the discussion;
  2. delete the content from P9L240 to 245 in the section of conculsion;
  3. change from present tense to past tense for all results which you got in MS.

Author Response

1 Response to comment:  We have strengthened each part of the discussion appropriately.

2 Response to comment: We delete the relevant content and modify the form and content of the conclusion. 

3 Response to comment: We have changed the tense for all results.

Reviewer 2 Report

The paper presents results of measurement and analysis of concentration of selected metals in campus area in Suzhou city in China. So the topic is of local importance. The paper can be published after major revision. The details are given below.

  1. The language must be improved. Most sentences are understandable, but the meaning of some can only be guessed. I suggest reading the paper by a native speaker.

  1. The term “heavy metals” is not recommended because it is not precise. Consider avoiding it.

  1. The introduction should be improved. The choice of metals should be explained.

  1. All positions and abbreviations in Table 1 should be explained. The description in lines 109-129 should be more precise. Consider adding one row in Table 1 corresponding to the sample number of the highest concentration. Besides, the table should be put after it is referred to in the text.

  1. Consider putting the formulas for Igeo and EF in separate lines.

  1. Lines 132-134 and 140-142: consider moving the text of 5 degrees to Subsection 2.2 near the definitions of Igeo and EF.

  1. Figure 2 - consider improving the figure by adding some lines facilitating the localization of the three clusters.

  1. Figure 3. The maps are too small to be legible. They must be enlarged. Consider using 3 larger maps per row.

  1. Every time “previous study” is used a relevant reference should be provided (e.g. line 205 or 218).

  1. Figure 4. The legend is too small.

  1. Lines 264-283 - complete and correct this fragment.

Author Response

1 Response to comment: We did our best to improved the language and grammar.

2 Response to comment: We replaced the term “heavy metals” in paper.

3 Response to comment: We added the explaintion for heavy metal selection and improved the introduction.

4 Response to comment: We explained all the abbreviations in Table1, and added the sample number of the highest concentration.

5 Response to comment: We used the formulas for Igeo and EF in separate lines.  

6 Response to comment: We moved the text of 5 degrees to section 2.2 in paper.

7 Response to comment: Figure 2 had been improved.

8 Response to comment: We modified Figure 3.

9 Response to comment: We have perfected the fragment in paper.

10 Response to comment: We modified Figure 4.

11 Response to comment: We corrected this fragment.

Reviewer 3 Report

The article addresses the very important problem of heavy metals accumulation in soils in urban areas. The authors tested 23 samples and 11 heavy metals in each sample. Besides, an attempt to determine the background for pollution and the degree of soil pollution was made. Geo-accumulation index (Igeo) and Enrichment factor (EF) methods were selected to assess the heavy metals pollution degree in soils. The work has an application character, but it contains many deficiencies that disqualify the paper.

  1. The work lacks reference conditions (isolated from anthropogenic factors), and repetitions at intervals. Without those, the authors can determine only instantaneous values of concentrations of individual metals while they are unable to determine the degree of pollution and trends in concentration distribution.
  2. Soil samples were taken from a depth level of 5-15 cm. In my opinion, it is not enough, and a second depth level should be determined to assess the potential leaching of elements deep into the soil profile.
  3. In the reviewed paper, I did not find data on pH levels in the analyzed samples. It is extremely important in the context of the general principle – while the pH value decreases, the solubility and mobility of heavy metals increases. In that way, the elements may be washed out into the deeper parts of the soil profile. This note is strongly linked to Note 2.
  4. The authors also do not specify the type of maintenance of the site (turf, field, street-side site, etc.). The way the soil is covered is an extremely important factor for the distribution and concentration of elements.
  5. The statistical methods presented in the Results section should be described in detail in the Material and Methods section.
  6. The discussion of the work should be based on the latest literature. To discuss the findings, only six entries in the reference list (references 15-20) were used by the authors.
  7. The references consist of only 20 entries: 7 of them are very old (from the 1990s or earlier), and only 6 are the latest publications – from the last decade. I suggest expanding the references to include the latest literature on the subject.

Author Response

1 Response to comment: The purpose of the study is to evaluate the concentration of heavy metals in soil and distinguish the source and background values. Therefore, although the work lacks reference, we believe that the instantaneous values of these metal concentrations can also have certain significance for the surrounding environment and spatial distribution.

2 Response to comment: The sampling is carried out in accordance with the relevant standards. Limited by the research purpose, we have not studied the soil profile, and will pay attention to it in the future research.

3 Response to comment: As in comment 2, the solubility and migration of heavy metal have not been studied due to the impact of research objectives, and will be paid more attention in the future research.

4 Response to comment: The relevant sampling point information is shown in Figure 1.

5 Response to comment: We added the description of the statistical methods in the section 2.2.

6 Response to comment: We supplement the literature and use it for discussion.

7 Response to comment: We have updated some references, but some of the old ones are original works, so we continue to quote them.

Reviewer 4 Report

The authors of the manuscript titled “Assessment, distribution and regional geochemical  baseline of heavy metals in soils of densely  populated area: a case study” evaluate the  concentrations of Cr, Mn, Co, Ni, Cu, Zn, As, Cd, Hg, Pb and Fe in 23 soil samples in Suzhou University, a typical densely populated areas. They also determine the geo-accumulation index and the enrichment factor. They find that Ni, Cu, As, Cd and Hg contents are higher than the background values of Chinese soil, and the CV of Hg and As are bigger. The authors individuate the causes of the increase or decrease of these heavy metals in this area and definite the background values of the heavy metals.

In my opinion,  this work was carefully conducted. The methods used are appropriate and the data interpretation is convincing. I believe the results are very interesting and original, therefore I strongly recommend the publication of this paper in International Journal of Environmental Research and Public Health after minor revision.

My observations are the following:

Although the paper is very interesting, the aspect of possible effects of some heavy metals in excess on human health is missing. Since the Journal is also focused on public health, I think that the authors must add these aspects in the introduction and / or discussion in light of recent works and describe also the importance of the new plant and animal bioindicators. I would also emphasize the effects of some of these excess metals on reproductive health which is critical for species survival.

For this aim I suggest reading and quoting the recent papers:

1. Selectivity of metal bioaccumulation and its relationship with glutathione S-transferase levels in gonadal and gill tissues of Mytilus galloprovincialis exposed to Ni (II), Cu (II) and Cd (II). Rend. Fis. Acc. Lincei 27, 737–748 (2016). https://doi.org/10.1007/s12210-016-0564-0

2. Relevance of  arginine residues in Cu(II)-induced DNA breakage and Proteinase K resistance of H1 histones. Sci Rep. 2018 May 9;8(1):7414. doi: 10.1038/s41598-018-25784-z

3. Mytilus galloprovincialis (Lamarck, 1819) spermatozoa: hsp70 expression and protamine-like protein property studies. Environ Sci Pollut Res Int. 2018 May;25(13):12957-12966. doi: 10.1007/s11356-018-1570-9

Author Response

According to the requirements, we added the description of the harm of heavy metals to human body, and referred to the relevant literature.

Round 2

Reviewer 2 Report

Most remarks have been taken into account. English still requires refinement (e.g. lines 38, 47, 52, 56 and others). The paper can be accepted after language corrections.

Reviewer 3 Report

I accept this paper in present form.